# A Comprehensive Review on Water Diffusion in Polymers Focusing on the Polymer–Metal Interface Combination

**DOI:** 10.3390/polym12010138

**Published:** 2020-01-06

**Authors:** Chao Yang, Xiao Xing, Zili Li, Shouxin Zhang

**Affiliations:** College of Pipeline and Civil Engineering, China University of Petroleum (East China), No. 66, West Changjiang Road, Huangdao District, Qingdao 266580, China; yangchao201001@163.com (C.Y.); xiaoxingupc@outlook.com (X.X.); lileixin@126.com (S.Z.)

**Keywords:** Fick’s diffusion, diffusion coefficient, relaxation, water clusters, permittivity, effective capacitance

## Abstract

Water diffusion in polymers is relevant to a broad range of physicochemical phenomena and technological processes. Although many fields contributed to rapid progress in the fundamental knowledge of water–polymer interactions, detailed understandings come mainly from interpreting numerous experiments. These studies showed that a remarkably rich variety of diffusion forms between water and even seemingly simple polymers. In this review, focusing on the gravimetric and capacitance method, we discuss contradictions and problems existing for water diffusion in polymers in detail from perspectives of experiments and models, focusing on the analysis of error derived from widely used methods, especially for the Brasher–Kingsbury equation. We also provide a perspective on outstanding problems, challenges, and open questions, including water clusters, relaxation, and electrochemical reactions at the metal/polymer interface, as well as expanding the theoretical prospective.

## 1. Introduction

To our best knowledge, organic polymers are widely used for protecting metal from corrosion. However, delamination of organic polymers is also a serious problem, especially caused by water diffusion, which attracted extensive attention [1]. This includes areas such as relaxation of organic matter, electrochemistry, and stress [2], to name just a few. Indeed, the ubiquitous presence of corrosive media in polymers under ambient conditions demonstrates that diffusion in polymers is relevant to many areas of the physical sciences. The water diffusion in polymers is widely studied in fields analyzing the failure of coatings caused by water and evaluating their protective performance. Furthermore, contemporary issues such as production technology and performance characterization in both self-healing coatings [3] and graphene coatings [4] indicate that there is a pressing need to better understand the dynamics of water diffusion in polymers. This, along with fundamental questions about hydrogen bonding and free energy, resulted in a flurry of interest in recent years, and the physicochemical characteristics of polymers in water represent one of the most exciting and thriving fields in materials chemistry and physics.

Water is a transmission medium of oxygen and ions, and water diffusion in polymers is a hot topic. The first step toward understanding water diffusion is examining diffusion path and relaxation conditions [5,6]. This is challenging because of the extreme complexity of polymer structure and composition [7], and the difficulty in analyzing the details of molecular-level explanation for such a small sample. Therefore, based on the number of assumptions, mathematical and electrochemical methods were proposed to describe the different processes of water diffusion in polymers. Fick’s second law puts forward the most direct description of this problem, and probably the biggest conceptual idea that emerged from well-defined studies of water diffusion in polymers is the Brasher–Kingsbury equation (B–K equation, for short).

The B–K equation is a concept used to discuss water diffusion in almost all polymers, and it is generally the first concept grasped by newcomers to this field [8,9]. Perhaps as a consequence, the B–K equation gained an importance that is entirely out of line with the experimental evidence in its favor, often being the last concept to be abandoned, only in the face of overwhelming evidence. The B–K equation is used in its strict sense, as well as more loosely, to discuss the change in water content in polymers over time, which was originally discussed on the basis of experiments in immersion, described by Fick’s second law. This review explores some of the groundbreaking new work published in this area recently, which calls this model for water diffusion into question. Indeed, here, we consider if the B–K equation was ever proven in the strict sense of the word.

According to the physicochemical parameters and the type of mathematical formulation used in the development of “microscopic” diffusion models, one could classify them as molecular, free volume, and hybrid models. This classification should, in principle, be valid for both rubbery and glassy polymers. However, there is a significant difference between the diffusion paths in rubber polymers (generation of micro-pores by fluctuation of polymer chains) and in glassy polymers (pre-existing micro-pores) [10]. Until now, more detailed and true “microscopic” treatments were mainly models for diffusion in rubbery polymers. An explanation for this may be the difficulty of developing consistent “microscopic” diffusion models for the much more complex diffusion process occurring in glassy polymers [11,12].

In this paper, as shown in Figure 1, focusing on epoxy resins, we explore gravimetric and capacitance methods, how they emerged, and how recent experiments and calculations revealed instead a much more interesting variety and richness of diffusion processes for water in polymers. The focus is on non-Fick’s diffusion, for which some of the most detailed and exciting recent experiments and models were performed. Key recent developments, including the derivation of calculated error in water content in polymers, are introduced. The challenges and issues that remain before a fully predictive understanding of water diffusion in polymers can be achieved are also discussed. This is a short review aimed mainly at newcomers to the field, using a few selected examples to illustrate the concepts and exciting developments alluded to above.

## 2. The Rise of Calculation Methods

Water diffusion in polymers is not only related to the protective coatings; it also plays an important role in the packaging of food and pharmaceuticals and inorganic–polymeric multilayer films for the encapsulation of flexible electronics. Here, many methods, e.g., carrier gas methods, were put forward to calculate the diffusion coefficient in different fields [13]. Among them, the gravimetric method (related to the intrinsic diffusion mechanisms in polymers) and capacitance method (related to the setting of the method) are widely accepted, and they are the research objects of this paper.

### 2.1. Gravimetric Method

Henry’s law only considers the diffusion process caused by the difference in water concentration, which can be described by Fick’s second law in a simple case, i.e., ideal Fick’s diffusion [14], for one-dimensional diffusion. Here, the diffusion in the normal direction is described as follows:(1)$$\frac{\partial c\left(x,t\right)}{\partial t}=D\left(\frac{\partial^{2}c\left(x,t\right)}{\partial x^{2}}\right),$$
where *c* is the concentration of water, *t* is the immersion time, and *D* is the diffusion coefficient, which is always assumed to be a constant. Typically, *D* is on the order of 10^−13^ m^2^/s, as reported in the literature [15,16]. For a polymer submersed at *t* = 0, the appropriate boundary conditions for *t* ≥ 0 are as follows:(2)$$c(x,0)=0,c\left(0,t\right)=c_{\infty },\frac{\partial c}{\partial x}\left(l,t\right)=0,$$
where *l* is the polymer thickness.

For the full solution of Equations (1) and (2) for water concentration *c*(*x,t*), see References [17,18]. The function *φ*(*t*), also known as the fractional mass *M*_t_/*M*_∞_, identified with the scaled average water concentration, is given below.
(3)$$\varphi \left(t\right)=\frac{M_{t}}{M_{\infty }}=1-\frac{8}{\pi^{2}}\sum_{n=0}^{\infty }\frac{1}{{\left(2n+1\right)}^{2}}\exp\left[\frac{-{\left(2n+1\right)}^{2}D\pi^{2}}{4l^{2}}t\right],$$
where *M*_t_ is the mass of water at time *t*, and *M*_∞_ is the mass of water in saturation.

Generally, the results at *n* = 10 match well with experiments, but the specific value of *n* should be determined using an actual model with an acceptable range of error [17,18].

### 2.2. B–K Equation or Capacitance Method

As described above, obtaining an accurate description of water diffusion in polymers is very challenging. Based on the development of electrochemical impedance, which can precisely reveal the changing physical properties of polymers caused by water diffusion, in 1954, Brasher and Kingsbury put forward a capacitance method for estimating the water uptake in supported polymers, roughly applicable to many polymer formulations, i.e., paint, coatings, and films, mostly agreeing with results obtained using the gravimetric method [19].

The original derivation begins with the law of mixtures proposed by Hartshorn et al. [20] for the relative permittivity (dielectric constant) of polymers.
(4)$$\varepsilon_{r}=\varepsilon_{c}^{V_{c}/V}\varepsilon_{w}^{V_{w}/V}\varepsilon_{a}^{V_{a}/V},$$
where *ε*_r_, *ε*_c_, *ε*_w_, and *ε*_a_ (*V*, *V*_c_, *V*_w_, and *V*_a_) represent the relative permittivity (volume) of wet polymers, dry polymers, water, and air, respectively. Considering *ε*_a_ = 1, Equation (4) can be written as follows:(5)$$\varepsilon_{r}=\varepsilon_{c}^{V_{c}/V}\varepsilon_{w}^{V_{w}/V}.$$

The capacitance of polymers can be described as follows:(6)$$C_{c}=\varepsilon_{r}\varepsilon_{0}\frac{A}{l},$$
where *C*_c_ is the polymer capacitance at time *t*, *ε*_0_ is the vacuum permittivity (*ε*_0_ = 8.85 × 10^−14^ F/cm), and *A* is the superficial area of polymers. 

Assuming that the volume of dry polymers is maintained during water diffusion, Equation (7) can be derived by combining Equations (5) and (6) as follows:(7)$$\frac{C_{c}}{C_{0}}=\frac{\varepsilon_{r}}{\varepsilon_{c}^{V_{c}/V}}=\varepsilon_{w}^{V_{w}/V},$$
where *C*_0_ is the dry polymer capacitance.

Then, by taking the log, an expression for *X*_V_ = *V*_w_/*V*, describing the water volume fraction in polymers, can be obtained, i.e., the normal B–K equation [19].
(8)$$X_{V}=\frac{V_{w}}{V}=\frac{\log(C_{c}/C_{0})}{\log(\varepsilon_{w})}.$$

By combining the gravimetric and capacitance methods, the amount of water taken up by polymers can be calculated from the capacitance values using the following relationship [21]:(9)$$M_{t}=\frac{V\rho_{w}}{\ln\varepsilon_{w}}\ln\frac{C_{c}}{C_{0}},$$
where *ρ*_w_ is the water density in polymers.

Assuming that *V*, *ρ*_w_, and *ε*_w_ are constant in the diffusion process, this equation is commonly employed in a normalized form as shown below [21].
(10)$$\frac{\ln C_{c}-\ln C_{0}}{\ln C_{\infty }-\ln C_{0}}=\frac{M_{t}}{M_{\infty }}.$$

Assuming an ideal Fick’s diffusion process, Equation (10) can approximately estimate the saturation time (the time at which the system reaches a specific capacitance or absorbs a certain mass of water), which is of great significance when determining the initial stage, as shown in Section 2.3. Under this assumption, a more accurate way to analyze the error of and relationship between these two methods is provided, especially when ascertaining the effective capacitance [1,3,21].

The capacitance method mainly relies on electrochemical measurements, which can obtain the capacitance of a whole system at different diffusion times, as shown in Equations (8)–(10). A three-electrode electrochemical test system is adopted, with the coated metal as the working electrode, a saturated calomel electrode as the reference electrode, and a platinum electrode as the auxiliary electrode. The electrochemical impedance spectroscopy (EIS) curves of the coated metal at different diffusion times are obtained, from which the capacitance of the whole polymer system can be extracted by equivalent circuit fitting. It should be noted that the fitted capacitance is the average capacitance value of the whole coating system, which is greatly affected by the electrical double layer at the polymer/metal interface [22].

### 2.3. The Signs of a Problem

As mentioned above, when assuming that water diffusion in polymers is regarded as an ideal Fick’s process, that is, water only diffuses in the micro-pores of polymers, then the process of water diffusion can be divided into three stages [23], which can be illustrated by the polymer capacitance over time, as shown in Figure 2.

(A) Homogeneous diffusion through micro-pores.

At this initial stage, Fick’s diffusion occurs, which is an ideal case of penetrant transport corresponding to free diffusion. Equation (3) can be used to describe this stage [17].
(11)$$\varphi \left(t\right)=\frac{2\sqrt{D}}{l\sqrt{\pi }}\cdot \sqrt{t}.$$

Combined with Equation (8), the relationship between polymer capacitance and diffusion time can be established as follows:(12)$$\frac{\log\left(C_{t}/C_{0}\right)}{\log\left(C_{\infty }/C_{0}\right)}=\frac{2\sqrt{D}}{l\sqrt{\pi }}\sqrt{t}.$$

(B) Saturation stage. At this stage, capacitance remains constant and the diffusion path is established.

(C): Electrochemical reaction. An electrochemical reaction occurs as water reaches the polymer/metal interface, and capacitance steadily increases due to the capacitance of the electrical double layer.

However, assumptions of an ideal process are too harsh, and they are always shown to be impractical by experimental phenomena, for instance, the incline or disappearance of a “platform” at the saturation stage. This phenomenon is tremendously dependent on the physical/chemical change of polymers, as detailed in the subsequent interpretation.

Figure 3 shows the error between experimental and theoretical values [24]. Based on the assumption of ideal Fick’s diffusion, at time *t* = 0–*t*_2_, the theoretical mass fraction of water (*φ*) shown in curve B agrees with Equation (10), that is, *n*_2_ = 0.5; once in the saturation state, *φ* is unchanged. However, at time *t* = 0–*t*_1_, shown in curve A, the *φ* obtained by experiment also satisfies Equation (10) with *n*_1_ > 0.5. For the second stage of saturation, it shows a slight increase, which is different from the results in curve B.

In most cases, the experimental results do not agree with ideal Fick’s diffusion. Based on the number of assumptions, related parameters were analyzed using gravimetric and capacitance methods, which can only give a preliminary explanation for water diffusion in polymers. However, it is difficult to clearly explain the diffusion process in micro-pores, let alone other interesting phenomena. Therefore, in order to clarify the mechanisms of polymer failure and metal corrosion induced by water diffusion, the effect of the assumptions on diffusion should be considered.

## 3. Effect of Polymer Characteristics on Water Diffusion (Gravimetric Method)

### 3.1. Glass Transition Temperature

The second assumption of ideal Fick’s diffusion is that the polymer variation caused by water diffusion can be negligible, which is absolutely not acceptable for most experiments. According to the relative rate of polymer relaxation compared to that of water diffusion, three types of water diffusion were illustrated by Frisch [25] as follows:(1)Case I (Fick’s diffusion, *n* = 0.5). Polymer relaxation is much faster than water diffusion, and the diffusion is followed by an instantaneous response of the system, resulting in Fick’s behavior. The instantaneous response of the system requires large flexibility of the polymer chains in the system, that is, the polymer is in a rubbery state. In this case, diffusion is controlled by the diffusion coefficient.(2)Case II (*n* = 1). The rate of diffusion is much faster than that of relaxation, which marks the innermost limit of water diffusion, and it is the boundary between the (stressed equilibrium) swollen gel and the glassy core of polymers. At this moment, the swelling of polymers occurs. In this case, the value of *n* is perhaps variable; for example, in Crank’s opinion, Case II is defined in the range of 0 < *n* < 1 [17].(3)Super Case II (Irregular diffusion, *n* > 1). Under these conditions, the rate of diffusion is equal to that of relaxation. The evolution, under certain circumstances, of Super Case II transformed from Case II makes it somewhat doubtful that this is as simple a limiting case as was once believed.

The relaxation rate of a polymer is closely related to its glass transition temperature (*T*_g_), which is defined as the temperature where the macromolecule segment movements become easier and more numerous than in the glassy state. Furthermore, when the macromolecular chains are less bonded to each other via chemical cross-linking nodes, they vibrate more easily under the effect of temperature; thus, the polymer network presents a lower *T*_g_ [1,3,13].

This rubbery state is generally accepted to be the case for polymers above their *T*_g_. In other words, if the experimental temperature is higher than the *T*_g_ of polymers, polymer chains can quickly return to their original state under external interference, illustrating that the water diffusion in polymers can be considered as ideal Fick’s diffusion [26]. On the other hand, the *T*_g_ reflects the curing degree of polymers, especially for epoxides, which is closely related to cross-linking density and free volume [27].

Considering that a considerable number of studies were carried out at a temperature much lower than the *T*_g_ of polymers, the above analysis can explain the phenomenon where *n* was slightly higher than 0.5 in the experiments [10,11,12,17,25].

However, a few scholars put forward that the irreversible interactions between water and polymers may occur at elevated temperatures, illustrating that the polymer backbone chains can be cut, and segments leach out during hygrothermal aging [28,29]. For instance, Xiao et al. [30] determined this phenomenon via measuring the water diffusion in DGEBA (Diglycidyl ether of bis phenol A) polymers (*T*_g_ = 14 °C) at 60 °C, showing that water could lead to the incorporation of C–O and N–CO–N groups, cutting the polymer backbone chains. In other epoxy resins, an interesting phenomenon occurred where there was a decrease in water content at elevated temperatures [31,32], which might be correlated to damage occurring in the cross-linked matrix induced by differential swelling stresses. It may also be explained by a thermodynamic argument, whereby the solvent combined with the polymer glass results in a state of internal equilibrium with an exothermic heat change and a volume contraction. An additional rationalization may involve the strength of hydrogen bonds between water and polymers, which is weakened as the temperature increases; subsequently, the amount of water bound to the network decreases [33].

The *T*_g_ (glass transition temperature) is a crucial factor affecting polymer properties, and there is not yet a clear explanation for this phenomenon. Two things should be noted about this transition. Firstly, its value depends on the speed of heating or cooling of the system. Secondly, water diffusion in polymers can lead to a change in *T*_g_, which presents the features of a total system with possible diffusion, meaning that it may transform from a glassy state to a rubbery state once the experimental temperature is near *T*_g_ [34].

Another potentially controversial question regarding *T*_g_ is whether curing time affects the cross-linking density of polymers. Macroscopic density was previously reported to decrease with a greater degree of cross-linking [35,36]. It was proposed that this occurs as a consequence of limited packing efficiency around cross-link sites [37,38]. An alternative explanation lies in the raised *T*_g_ of more highly cross-linked samples, meaning that these are further from their equilibrium conformation when the structure is frozen upon cooling [39]. A recent report on cure time and free volume for epoxy amine systems found no difference between samples cured for different times, indicating that the relationship between free volume and cure time might be system- or cure-schedule-dependent [40]. At present, however, it appears that extremely small chemical changes occurring during a prolonged cure (i.e., the consumption of residual epoxy groups not detectable by Fourier Transform Infra-Red) have a significant effect on the physical properties [41]. This issue is perhaps related to the nature of polymers, as continuously discussed and explored in various studies.

### 3.2. Effect of Experimental Temperature on Diffusion Coefficient

Several studies were carried out at ambient temperature, as rising experimental temperature may deform the physical structure, thereby enlarging or contracting the free volume, instead of damaging of chemical chains of the polymers [36]. This explanation can be supported by Castela’s experiments [42]. Only polymer-based paint was exposed to 125 °C (*T*_g_ = 8 °C) for a long time, damage to the chemical chains occurred.

The correlation between diffusion coefficient (*D*) and temperature can be theoretically illustrated by the Arrhenius equation, which features two assumptions: (1) water diffusion has no influence on the physical properties of polymers; (2) the activation energy (*E*_a_) of a diffusion process is independent of temperature [43,44].
(13)$$D=D_{0}\exp\left(\frac{-E_{a}}{RT}\right),$$
where *D*_0_ is the theoretical diffusion coefficient at infinite time [45], *E*_a_ is the activation energy, *R* is the gas constant, and *T* is the absolute temperature.

Based on the assumptions of the diffusion process only being influenced by the experimental temperature and of a uniform temperature distribution in polymers, a fitted curve of ln*D*~(1/*T*) can be used to extract *D*_0_ and *E*_a_, thereby discovering an “unmatched” result compared to the above two assumptions [46]. Actually, the assumption of activation energy and temperature was not always valid in numerous experiments [47,48]. A perfect linear curve of ln*D*~(1/*T*) cannot be obtained. The main cause of this error is ignoring the change in activation energy due to the unexpected interactions at different temperatures.

On the basis of Equation (13), other approaches can be used to acquire an acceptable activation energy, instead of the diffusion coefficient. Thus, the difference between diffusion coefficients mainly becomes a function of temperature, for which Equation (14) is commonly used in experiments with values obtained for the diffusion coefficient (*D*) at two different temperatures [24].
(14)$$\ln\frac{D_{1}}{D_{2}}=\frac{E_{a}}{R}\left(\frac{1}{T_{2}}-\frac{1}{T_{1}}\right).$$

However, the above equation relies on experimental methods, such as the gravimetric method and the carrier gas method, to obtain the diffusion coefficient, which undoubtedly considers the assumption of ideal Fick diffusion, especially with regard to the assumption of error in calculating the diffusion coefficient.

### 3.3. Diffusion Coefficient and Solubility Coefficient

The above discussion focused on the physical structure or free volume affected by absorbed water, not taking into account the interaction between water and polymers, which is another main error when calculating the diffusion coefficient. In 1989, Scully [49] explored the required time for the diffusion of oxygen through polymers to a coated metal interface, where oxygen was electrochemically reduced, which was given by the following expression using Fick’s second law:(15)$$t_{s}=0.0653\frac{l^{2}}{D},$$
where *t*_s_ is the saturation time, meaning water reached the polymer/metal interface. The constant 0.0653 corresponds to a condition where the diffusion flux at the coated metal interface is one-tenth of the final steady-state diffusion flux, assuming a sufficient reaction of oxygen at the coated metal.

In 2004, Equation (15) was proven by Hu’s experiment [50], illustrating that the use of this simple expression permits a direct comparison of the time required for the initial diffusion of each species to the coated metal interface. For instance, in Hu’s experiment using an NaCl solution diffusing in an LY12-aluminum system, the calculated saturation time (*t*_s_) was 44.6 min with *l* = 10^−2^ cm and *D* = 2.44 × 10^−9^ cm^2^/s, while the experimental value was 50 min.

Another simplified equation for optimizing preferred polymers was put forward in 1996 by Diguet [51].
(16)$$D_{\mathrm{app}}=\frac{{\left(2l\right)}^{2}}{t_{s}},$$
where *D*_app_ is the “apparent” diffusion coefficient.

The above analysis illustrates an ideal Fick’s diffusion process, giving rise to a diffusion coefficient (*D*), which can approximately represent the total rate. However, it is impossible to ignore the interaction between water and polymers, which leads to the main error shown in Figure 3. The nonlinear sorption, resulting in a deviation from ideal Fick’s law, is the interaction of water with or absorption of water by polymers, which may cause the rearrangement of chemical chains; this process is described by the solubility coefficient (*S*). Therefore, the permeability coefficient (*P*), expressing the diffusion rate caused by the interaction between water and polymers, can be described as follows:(17)P=D×S.

A similar equation should be applied to quantify the solubility [40].
(18)$$S=S_{0}\exp\left(\frac{-\Delta H}{RT}\right),$$
where Δ*H* is the enthalpy of sorption (heating of solution), and *S*_0_ is the constant pre-exponential coefficient. 

Subsequently, the permeability exhibits an Arrhenius relationship,
(19)$$P=P_{0}\exp\left(\frac{-E_{p}}{RT}\right)m$$
where *E*_p_ = *E*_a_ + Δ*H* is the activation energy for permeability, and *P*_0_ = *D*_0_·*S*_0_ is the constant pre-exponential coefficient.

Many studies suggested that the order of magnitude of total “diffusion” coefficient (*D* and *P*) in different polymers was 10^−11^–10^−7^ cm^2^/s [52,53]. Coniglio [49] investigated the water sorption kinetics and equilibrium in 100% solid epoxy coatings through their immersion into distilled water baths from 20 to 85 °C (absorption tests) and drying in a furnace between 60 and 85 °C (desorption tests), in which the diffusion coefficient, solubility, and permeability were calculated. The results showed that permeability was about 1%–3% of the diffusion coefficient (Table 1), indicating that the assumptions in the above analysis are acceptable.

However, in some special polymers, the result largely deviates from an ideal process due to the absorption of water by polymers, indicating that water may prefer to choose special groups in the polymers, leading to a rearrangement of chemical chains [55].

It should be noted that the solubility coefficient (*S*) decreases with the increase in temperature, indicating that there may be an elevated temperature at which permeation does not occur. This critical temperature may be the glass transition temperature (*T*_g_) of polymers.

Another amazing idea was explored by scholars. Theoretically, the interaction between water and functional groups in polymers always takes place after water enters the polymers. Thus, how long after water enters the polymers does the interaction take place? Alternatively, at what stage does ideal Fick diffusion happen?

### 3.4. A Special Stage to Describe the Ideal Fick’s Process

In theory, micro-pores are preferred diffusion paths in polymers where interactions may occur because of their lower resistance [56]. Based on this assumption, the calculation of diffusion coefficients in the initial period can be greatly simplified. Therefore, Philippe [57] proved this case for the diffusion process in a short time of *t* = 20–60 min using EIS and ATR-IR (attenuated total reflection infrared spectroscopy) applied to commercial epoxy polymers. Here, once *t* > 1 h, the interaction and the variable polymer thickness were non-negligible, which was identified as the initial stage. Similarly, other scholars [58] defined the initial diffusion stage satisfying ideal Fick’s law, which was defined as, for example, *M*_t_/*M*_∞_ < 0.6, by Miwa using a CCT (cyclic corrosion test) [59,60].

Vrentas [61] introduced a diffusion Deborah number (DEB)_D_ to indicate the presence of non-Fick effects during absorption experiments. This number is defined as follows:(20)$${\left(\mathrm{DEB}\right)}_{D}=\frac{\lambda_{m}}{\theta_{D}},$$
where *θ*_D_ is the charecteristic time of penetrant diffusion, and *λ*_m_ can be considered as the characteristic time of the polymer relaxation preocess. *θ*_D_ is proportional to the square of film thickness *l*; consequently, (DEB)_D_ is proportional to *l*^−2^.

*T*_E_ is the temperature below which pure polymers act as an elastic solid, *T*_g_ is the glass transition temperature, and *T*_V_ is the temperature above which pure polymers act as a viscous fluid. Vrentas [61] claimed that Fick’s behavior was found both above *T*_V_ and below *T*_E_. In the latter state, the polymer segments instantaneously move back to their original positions after the passage of a diffusion particle, resulting in a net unchanged matrix. In our opinion, rigid structures should not be excluded from this point of view. As shown in Figure 4, the (effective) glass transition temperature is in the area between these borderlines; thus, in this respect, Vrentas disagrees with the general opinion that the glass transition temperature is the border between Fick’s and non-Fick’s diffusion characteristics [1].

Temperature is the most important factor affecting polymer morphology, which is related to the ability of water to interact with polymers, dependent on functional groups. Until now, this question was continuously the focus of discussion and debate. A method to determine whether water and functional groups can interact or not is the key to clearly discerning these differences.

### 3.5. A Method to Discern the Different Processes of Water Diffusion

As is well known, it is impossible to clearly distinguish the process of water diffusion in micro-pores or its interaction with functional groups, whereas the “diffusion rate” in variable processes shows enormous differences. To analyze in more detail the water uptake kinetics, the evolution of the electrolyte flux with immersion time was evaluated via differentiation of the exponential expression, raised by Pérez et al. [24] as follows:(21)$$\varphi =kt^{n}\Rightarrow \mathrm{Electrolyte}\;\mathrm{flux}=\frac{d\varphi }{dt}=n\cdot k\cdot t^{n-1}.$$

As expected, the electrolyte flux decays with the immersion time. In ideal Fick’s diffusion, there is only one time constant (*τ*) for the period of water diffusion, which may only take place for some special polymers, i.e., rubber. In many cases, several time constants represent different diffusion stages, especially for water-borne polymers [24].

Although the physical interpretation of these time constants remains unclear, several researchers took steps toward a better understanding of the non-ideal diffusion process.

However, *τ* is the comprehensive result of a different process. That is to say, it does not necessarily represent a real diffusion process. Wind et al. [14] proposed that the water diffusion process in polymers mainly considers two aspects: (1) a structural parameter related to the non-uniformity of polymers, and (2) a dynamic parameter related to relaxation. These aspects were represented by different interaction processes and their associated relaxation constants. A phenomenological model was developed to reflect the different rates of water diffusion in more clearly established polymers due to the polymers not being a continuum medium [14].
(22)$$\varphi_{\left(t\right)}=A_{1}\varphi_{0}\left(t/\tau_{1}\right)+A_{2}\varphi_{0}\left(t/\tau_{2}\right),$$
with
(23)$$\varphi_{0}=1-\frac{8}{\pi^{2}}\sum_{n=0}^{\infty }\frac{1}{{\left(2n+1\right)}^{2}}\exp\left(\frac{-{\left(2n+1\right)}^{2}D\pi^{2}}{l^{2}}t\right),$$
where *φ*_(*t*)_ is the saturation function, which is the reduced volume fraction of water uptake, *φ*_0_ is the solution of the *φ*_(*t*)_ function supposed to be ideal Fick’s diffusion, *A*_1_ and *A*_2_ are standardized parameters where *A*_1_ + *A*_2_ = 1, *τ*_1_ and *τ*_2_ are the time constants associated with different diffusion rates, and *τ* = *l*^2^/*D*.

Frisch [25] emphasized the dynamic factors, including rearrangements in the polymer structure, as a response to the sorption and diffusion process of water, assumed to be responsible for deviations from ideal Fick’s behavior. These structure changes have corresponding relaxation time constants. In addition, Berens et al. [62] developed a relaxation model based on this idea (Equation (23)) with two different contributions: a fast one (*τ*_D_) related to the diffusion process of the electrolyte across the polymers, and a slow one (*τ*_R_) describing the relaxation phenomena associated with the polymer rearrangements.
(24)$$\varphi_{\left(t\right)}=A_{D}\varphi_{0}\left(t/\tau_{D}\right)+A_{R}\left(1-\exp\left(-t/\tau_{R}\right)\right).$$

This similar expression indicates that the cross-linking structure inside polymers is a key factor affecting the diffusion rate of water, such as voids or low cross-link zones where the diffusion rate is much higher and, hence, the associated time is smaller; on the other hand, the most compact zones have a lower diffusion rate and, as a consequence, a higher time constant. Pérez et al. [24] proved that the number of types of functional groups with significantly different characteristics is equal to the number of time constants, which is merely a guide for a detailed description of the diffusion process, but which makes no sense due to the complexity of polymers.

## 4. Errors Deviating from B–K Equation

Errors of the capacitance method when calculating the water content in polymers (B–K equation) are derived from three aspects: polymer thickness, water permittivity, and electrochemical reactions at the polymer/metal interface. Most of all, according to the use of EIS to determine the capacitance, the capacitance of an electrical double layer at the polymer/metal interface dominates the accuracy. These phenomena are discussed below.

### 4.1. Variable Polymer Thickness with Time

In the assumption of ideal Fick’s diffusion, an obvious factor leading to errors between experimental and theoretical results is the change in polymer thickness caused by water diffusion. As mentioned above, ideal Fick’s diffusion conclusively demands a faster response of relaxation. Under this circumstance, the effect of water diffusion on the polymer’s physical/chemical properties is negligible. Several studies proved that the process of water diffusion in rubbery polymers was approximately considered as Fick’s diffusion within acceptable error. However, glassy polymers cannot meet the above conditions, with regard to a change in polymer thickness, swelling, or shrinking.

To our best knowledge, the change in polymer thickness is affected by many factors, and the main causes include the following two aspects:(1)The volume of micro-pores is indirectly changed due to water diffusion, as well as the formation of water clusters, which both deform the physical structure of polymers [1,5].(2)The rearrangement of polymer chains induced by the relaxation process may change the chemical structure, which is also referred to as the swelling phenomenon [6,21,28,29,30,31].

However, Jelinski et al. [63] reached different conclusions using quadrupole echo deuterium NMR spectroscopy. Their results revealed that (a) water in epoxy resin was impeded in its movement, (b) there was no free water, (c) there was no evidence for tightly bound water, and (d) it was unlikely that the water disrupted the hydrogen-bonded network in the epoxy resin. The water molecules migrated from site to site, but the jumping motion did not involve a specific hydrogen-exchange mechanism. Studies using dielectric experiments also suggested that the water was not bound to functional groups in the resin or to hydrogen-bonding sites [5]. There was only some clustering of water molecules in the polymers, rather than complete molecular separation.

Numerous similar experiments obtained different conclusions and deductions. What are the controversial points among the numerous conclusions, and what are the main aspects causing the change in polymer thickness?

#### 4.1.1. Water Clusters

The swelling or shrinking polymer thickness is the result of a variable internal structure, especially in terms of the physical characteristics. An experiment to discern whether an abnormal interaction of water occurred or not was carried out by Bouvet et al. [64], exploring deionized water diffusion in a free film of DGEBA (Diglycidyl ether of bis phenol A) poly epoxide resin at 30 °C, as shown in Figure 5.

The whole period of water diffusion could be divided into three stages.

In stage I, the swelling volume of polymers (Δ*V*) was equal to the volume of water entering the polymers (*V*_w_), approximately denoting that the volume of the whole system consisted of water and polymers without air or free volume at *V*_w_/*V*_c_ = 1.3%. Several researchers probably considered this stage as demonstrating ideal Fick’s diffusion, contradictory to Bouvet’s explanation [64], whereby the swelling was caused by the rupture of hydrogen bonds. However, the above explanation was proven to be right by Ruller et al. [65] using nuclear magnetic resonance, and a similar deduction was drawn by Miwa et al. [59], who showed that the diffusion coefficient of absorption was lower than that of desorption, noting that, in the process of water diffusion, the water molecules need to overcome the obstruction of chemical chains in polymers.

Stage II occurred in the range of *V*_w_/*V*_c_ = 1.3%–3%. It was interesting that the curve of Δ*V* was lower than the curve of *V*_w_, suggesting the formation of water clusters [66,67], meaning that *n* individual water molecules gathered and combined to form larger water groups. This verified that the volume occupied by a cluster of *n* water molecules was lower than that occupied by *n* individual water molecules. 

Stage III lay in the range of *V*_w_/*V*_c_ > 3%, presenting a linear relationship of Δ*V* and *V*_w_, in which the slope was the same as that in the first stage. This phenomenon could be explained by the water clusters reaching their maximal size. Moreover, the formation of water clusters required a sufficient number of water molecules, which could not occur in the initial stage.

Musto et al. [68,69], and Wu and Siesler [70] reported a slight decrease in the ratio of free to bound water with increasing moisture content. However, in the case of an epoxy–phenolic network studied using 2D-ATR (two-dimensional attenuated total reflection), Lui et al. [71] concluded that the absorption of bound water occurred prior to diffusion into the bulk. It is probable that the characteristics of water uptake are particular to the polymeric system studied, although an alternative explanation for this ambiguity lies in the ATR methodology used in the latter study to monitor water uptake.

#### 4.1.2. Water Absorption and Desorption in Polymers

In Section 4.1.1, the absorption preferentially featured interactions with H, in both water and polymers. Surprisingly, in Antoon’s experiment [72], a completely reversible phenomenon was found in the interaction between water and epoxy using Fourier-transform infrared spectroscopy (FTIR).

However, the theory of hydrogen bonding was rejected by Jelinski et al. [63], while additional experiments also suggested that the water was not bound to functional groups in the resin or to hydrogen-bonding sites. There was only some clustering of water molecules in polymers, rather than complete molecular separation [25].

Coniglio et al. [54] measured the absorption and desorption of water in 100% solid epoxy coatings, which showed an amazing result of complete desorption. However, alternating sorption–desorption cycles induced coating degradation, illustrating that the absorption and desorption of water had little effect on the physical and chemical properties of the polymers.

Different polymers feature different numbers of functional groups and a unique free volume, leading to variable characteristics in water diffusion due to their diversity of distribution and type. An increasing number of studies analyzed polymer swelling caused by water diffusion, and the problems undoubtedly correspond to two aspects. One is the formation condition of water clusters, and the other is based on whether or not the interaction between water and polymers occurs (as shown in Figure 6 [73]).

This phenomenon is perhaps related to the hydrogen-bond energy and activation energy; that is to say, the relationship between these two values is the key to the formation of hydrogen bonds. However, it is important to note that temperature is a very important factor in this research, which not only affects the energy needed to power hydrogen-bond formation, but also significantly influences the nature of polymers. Accurately understanding the relationship among polymer characteristics, physical structure, and chemical properties is of great significance for revealing the process of water diffusion [74].

#### 4.1.3. A Parameter for Describing Variable Polymer Thickness in the Gravimetric Method

However, because of the complexity of polymers, the uncertainty of which functional groups interact with water, the forms of rearrangement, etc., it is difficult to explain the swelling phenomenon from the aspect of a change in physical/chemical structure. As a consequence, based on Equation (3), in a water–epoxy system, van Westing et al. [56] put forward a swelling coefficient (SC_m_) to describe the further absorption of water in water–epoxy systems induced by the swelling phenomenon.
(25)$$M_{t}=M_{\infty }\left(1-\frac{8}{\pi^{2}}\sum_{n=0}^{\infty }\frac{1}{{\left(2n+1\right)}^{2}}\exp\left[\frac{-{\left(2n+1\right)}^{2}D\pi^{2}}{4l^{2}}t\right]\right)+{\mathrm{SC}}_{m}\cdot t,$$
where *M*_t_ is the mass of water at time *t*, *M*_∞_ is the absorbed mass at infinite time originating from Fick’s diffusion, SC_m_ is the swelling coefficient obtained from mass measurements, and *l* is the thickness of supported polymers.

Combining with Equation (9), for the capacitance of polymers, the equivalent formula can be defined as follows:(26)$$\log\left(C_{t}\right)=\log\left(\frac{C_{s}}{C_{0}}\right)\left(1-\frac{8}{\pi^{2}}\sum_{n=0}^{\infty }\frac{1}{{\left(2n+1\right)}^{2}}\exp\left[\frac{-{\left(2n+1\right)}^{2}D\pi^{2}}{4l^{2}}t\right]\right)+{\mathrm{SC}}_{c}\cdot t+\log\left(C_{0}\right),$$
where *C*_t_ is the capacitance of polymers at time *t*, *C*_s_ is the capacitance at infinite time originating from Fick’s diffusion, *C*_0_ is the capacitance of polymers at time *t* = 0, and SC_c_ is the swelling coefficient obtained from capacitance measurements.

As mentioned above, some scholars defined the initial stage of water diffusion in polymers, suggesting that it obeyed Fick’s law, while others proved that diffusion and relaxation were simultaneous, whereby *n*_1_ > *n*_2_ and *t*_1_ > *t*_2_, as shown in Figure 3. The collision of these two ideas makes it impossible to pinpoint a physical meaning of the swelling coefficient (SC_m_ or SC_c_). In our opinion, the swelling coefficient (SC_m_) is different from the mechanism of simple polymer rearrangement of organic matter with water in terms of the two following aspects:(1)In long-term gravimetric experiments, the form and distribution of water in polymers may change. The swelling phenomenon of polymers may only be caused by a change in physical structure without a chemical reaction [59].(2)There may be two processes of water behavior, i.e., a combination with or precipitation from polymers, which may result in a reduction in thickness [6].

In our opinion, the physical interpretation of SC is not so simple, as mechanisms other than those related to rearrangements in polymer structure or swelling can induce a change in capacitance during long exposure times, such as (i) the slow diffusion of ions that likely follows water absorption [75], (ii) the partial dissolution and leaching out of organic fragments (revealed by gravimetric measurements), and (iii) a change in the distribution and/or state of water, which is also likely to occur during long exposure times.

### 4.2. Permittivity

The errors caused by permittivity in the B–K equation mainly consider three factors, including ion diffusion, an increase in water volume, and individual or clustered water in polymers, as discussed below.

#### 4.2.1. The Process of Ion Diffusion

In EIS, ions are always added to deionized water to promote the conductivity of the whole system, thereby easily obtaining the electrochemical signal. However, this practice leads to the introduction of another diffusion process, changing the permittivity used to calculate the capacitance via Equation (8) [76,77].

#### 4.2.2. Increasing Water Volume in Polymers

Water diffuses into polymers to supersede the air, leading to an increase in *V*_w_/*V* in Equation (4), thus predicting the increase in permittivity of the whole system. For example, in the case of a 3 wt.% NaCl solution entering epoxy polymers, the saturated water volume fraction can reach about 5%. According to the assumptions of water permittivity (*ε*_w_ = 80) and dry polymer permittivity (*ε*_c_ = 8), the relative change rate of permittivity of wet polymers with respect to dry polymers is 11.6%, implying an unacceptable error using the B–K equation, even though the capacitance can be easily obtained through EIS.

In order to consider the influence of water diffusion on the capacitance of the whole system, it is assumed that the diffusion is uniform at any time (see Figure 7). According to Equation (27), the capacitance of the whole system at time *t* can be estimated [78].
(27)$$\frac{1}{C_{c}}=\frac{1}{C_{W}}+\frac{1}{C_{D}}=\frac{l_{1}}{\varepsilon_{r1}\varepsilon_{0}A}+\frac{l_{2}}{\varepsilon_{r2}\varepsilon_{0}A}=\frac{1}{\varepsilon_{c}^{\alpha }\varepsilon_{0}A}\left(\frac{l_{1}}{\varepsilon_{w}^{\beta }}+\frac{l_{2}}{\varepsilon_{a}^{\beta }}\right),$$
where *ε*_r1_ and *ε*_r2_ represent the permittivity of dry and wet polymers, respectively, *l*_1_ and *l*_2_ represent the thicknesses of dry and wet polymers, respectively, and *α* and *β* represent the volume fractions of dry polymers and water/air, respectively.

The determination of *β* in Equation (27), representing the volume fraction of water/air in polymers, can be calculated using the isobaric adsorption of N_2_, or it can be calculated using Equation (28) [79].

When water in polymers reaches saturation, the water volume is the sum of the volume occupied by water/air.
(28)$$\varphi_{\infty }=\frac{\rho_{c}\cdot M_{c}}{\rho_{w}\cdot M_{\infty }}=\beta ,$$
where *φ*_∞_ is the saturated volume fraction of water, *ρ*_w_ and *ρ*_c_ are the densities of water and dry polymers, respectively, and *M*_c_ and *M*_∞_ represent the mass of dry polymers and water in saturation, respectively.

The above model is similar to the discrete model (DM), put forward by Bellucci et al. [80], which is one of the most important studies on the prediction of water content, in addition to the continuous model (CM) (Figure 8).

DM model: According to this model, it is assumed that the water concentration in the homogeneous polymers is constant at any time *t* between zero and saturation. Water front variations occur, in this case, and the water concentration is a function of time *t*, not of the polymer thickness. The equivalent analogue can be assumed using a parallel RC (resistance-capacitance) circuit, as shown in Equations (29) and (30) [80].
(29)$$\frac{C_{t}}{C_{0}}=1+2.3\log\left(\varepsilon_{w}\right)\frac{K\varphi_{\infty }}{M_{\infty }}t^{n}.$$
(30)$$K=\frac{4M_{\infty }\cdot D^{0.5}}{\pi^{0.5}\cdot l}\left(n=0.5\right).$$

Equation (29) is valid only for short times, and it can be used to evaluate both the exponent *n* and the initial film capacitance. Contrary to the gravimetric method, a nonlinear regression analysis must be applied in this case to the experimental capacitance data to obtain the values of both *n* and *C*_0_. Note that, at this stage, the water diffusion coefficient and the saturated water fraction cannot yet be calculated. The diffusion coefficient *D* can be evaluated for a Fickian diffusion phenomenon (*n* = 0.5) according to the analysis reported below.

CM model: According to this model, the film is divided into layers of polymer thickness. Due to the water concentration across the polymers, each layer contributes to the overall film capacitance *C*_t_ with its own value.
(31)$$\frac{1}{C_{t}}=\frac{1}{C_{0}}\left[1-\frac{2.3\log\left(\varepsilon_{w}\right)}{\rho Al}Kt^{n}\right],$$
where *A* is the area, and *l* is the thickness of experimental polymers.

Equation (31) can be used to evaluate both the exponent *n* and the initial polymer capacitance *C*_0_ through a nonlinear regression analysis of the experimental capacitance data as 1/*C*_t_ ~ *t*^n^. At this stage, only the mechanism of water transport across the polymers can be determined. Contrary to the DM model, the diffusion coefficient (*D*) can be extracted from the capacitance time decay plot only when the mechanism is Fick’s diffusion and for small values of the saturated water fraction (*φ*_∞_ << 1).

#### 4.2.3. Individual or Clustered Water

A controversial issue concerning water diffusion in polymers is that, once a certain amount of water accumulates, the individual water molecules may gather to form water clusters, or they may combine with functional groups in polymers. However, there is no doubt that both of these aspects undoubtedly cause the water permittivity to deviate from 80, as mostly accepted in the B–K equation. The former possibility may change the water permittivity from 80 to 60 when free water becomes bound, whereas the latter possibility may change the local temperature in the polymers, thereby affecting the local permittivity of water.

Based on the theory of Hartshorn et al. [20], the relationships most commonly used to calculate the water uptake from impedance data are given below.

Considering the content of water molecules in the coating as far lower than 1, the permittivity changes linearly with the volumetric moisture content, and the linear relationship is as follows:(32)$$\varepsilon_{r}=\varepsilon_{c}\left(1-\varphi \right)+\varepsilon_{w}\varphi .$$

The logarithm of both sides of Equation (4) [20] can be taken to obtain Brasher and Kingsbury’s formula [14].
(33)$$\log\left(\varepsilon_{r}\right)=\log\left(\varepsilon_{c}\right)+\log\left(\varepsilon_{w}\right)\varphi .$$

Sykes’s formula [81] introduced the effect of polymers on water diffusion.
(34)$$\log\left(\varepsilon_{r}\right)=\log\left(\varepsilon_{c}\right)\left(1-\varphi \right)+\log\left(\varepsilon_{w}\right)\varphi .$$

Nguyen et al. [82] analyzed the influence of the expression form of permittivity compared with the gravimetric result, suggesting that the linear relationship (Equation (32)) may be more or less adequate for different systems, while unacceptable error occurred when using the Sykes formula (Equation (34)). This is mainly because Sykes overestimated the influence of polymers on the water diffusion.
(35)$$\varphi =\frac{V_{w}}{V}=\frac{\log(C_{c}/C_{0})}{\log(\varepsilon_{w}/\varepsilon_{c})}.$$

The denominator in the B–K equation (Equation (8)) is log(80). For a polymer with *ε*_c_ = 3, this becomes log(26.7) in the new equation. This increases the calculated water fraction by a factor of roughly 1.43, which is a significant difference. Given the good empirical correlation found in practice between the B–K equation and gravimetric results, there seems to be no useful purpose in advocating that this variant be used instead; deviations from the gravimetric results are often in the opposite direction. It is useful, however, to be aware of the built-in approximation in their equation.

### 4.3. Effective Capacitance

In the capacitance method, equivalent electric circuits (EECs) [83] are used to determine the capacitance of polymers; then, based on Equation (5), the volume fraction of water in polymers can be calculated. However, due to the porous nature and surface inhomogeneity of polymers [84], the constant phase angle component (*CPE*) is usually used to replace pure capacitance (*C*) in equivalent electric circuits. The resistance of *CPE* can be mostly calculated as follows [85,86]:(36)$$Z=\frac{1}{Y_{0}}{\left(jw\right)}^{-n},$$
where *Y*_0_ is the capacitance of *CPE*, and *n* is the dispersion index.

To our best knowledge, the results obtained from the conventional EIS method reflect the “average” characteristics of the entire test electrode, which describes the current density distribution on the surface in a specific range of frequency [87,88]. However, once the diffusion path is formed in polymers, the non-Faradaic current in these paths is much greater than the Faradaic current, which can severely affect the accuracy of EIS curves. Therefore, the equivalent capacitance was put forward to eliminate the error.

At the range of frequency in EIS, the charge transfer process occurs at high frequency, while the mass transfer process occurs at low frequency. Generally, at the highest frequency of *f* = 10 kHz, the capacitance of polymers (*C*_c_) is mainly affected by charge transfer rather than mass transfer, which is called *C*_HF_, calculated as follows [89,90]:(37)$$C_{\mathrm{HF}}=\frac{-Z^{\prime \prime }}{2\pi f\left(Z^{\prime }{}^{2}+Z^{\prime \prime }{}^{2}\right)},$$
where *Z*′ and *Z*″ are the real and imaginary resistances at *f* = 10 kHz, respectively.

Considered as a global variable for describing the whole diffusion process, the effective capacitance of *C*_effect_ was put forward to calculate the volume fraction of water in polymers. Hence, Brug et al. [91] and Hsu et al. [92] came up with equations for calculating *C*_effect_ from the perspective of tangential and normal distributions of the time constant, respectively, as shown in Figure 9. Brug et al. [91] verified that the solution resistance is not negligible in the analysis of CPE behavior for a time constant on the surface, and *C*_effect_ can be calculated as follows:(38)$$C_{\mathrm{Brug}}=Y_{0}^{1/n}{\left(\frac{1}{R_{s}}\right)}^{\left(n-1\right)/n}.$$

On the contrary, a normal distribution of the time constant was assumed by Hsu et al. [92], whereby the total resistance contains the characteristics of the whole system. Based on this assumption of normal distribution, solution resistance can be ignored, and *C*_effect_ can be calculated as follows:(39)$$C_{\mathrm{Hsu}}=Y_{0}^{1/n}{\left(\frac{1}{R_{c}}\right)}^{\left(n-1\right)/n}.$$

Based on the B–K equation, Nguyen et al. calculated the volume fraction of water using *C*_HF_, *CPE*, and *C*_effect_. The results showed that the *X*_V_ obtained using *C*_effect_ agreed with the result of the gravimetric method, while the others had huge errors. The results obtained from three values of *X*_V_ can be described as follows [22]:(40)$$X_{V}\left(\mathrm{CPE}\right)\approx 1.5X_{V}\left(C_{\mathrm{HF}}\right)\approx 1.5^{2}X_{V}\left(C_{\mathrm{Brug}}\right).$$

### 4.4. Electrochemical Reaction at Polymer/Metal Interface

From the aspect of thermodynamics, stronger hydrogen bonds are easily formed by a combination of water and oxygen functional groups in polymers, which may destroy the weaker bonds of polymers and metal, resulting in delamination. Furthermore, water reaching the interface forms an electrolyte macroscopic phase, which changes the nature of the metal surface. Under this circumstance, a corrosion cell is gradually formed, which provides continuous conditions for cathodic delamination [87]. Therefore, in early studies, it was considered that, as long as water reached the interface, polymer delamination would occur due to the destruction of combined bonds of polymers and metal [93,94]. However, Nazarov et al. [95] proved that the influence of water was reversible, whereby, when water left to form a stronger hydrogen bond, the bond between polymers and metal reverted to its original state. Another experiment showed that, even though water reached the interface, it did not result in a decrease in binding force. Leng et al. [96] studied the accumulation of water at the polymer/metal interface using FTIR, as well as the adhesion strength between the polymer and metal using a mechanical device. The results showed that water could quickly penetrate into the polymers and gathered at the interface. However, even if the concentration at the interface was five times greater than that in polymers, there was no reduction in binding force between the polymer and metal, which ruled out an individual effect of water on delamination [97]. Therefore, it can be concluded from the above analysis that, as water reaches the polymer/metal interface, an electrochemical reaction is a sufficient condition for polymer delamination. That is to say, O_2_ + 2H_2_O + 4e^−^→4OH^−^ occurs at the cathode, and Fe→Fe^2+^ + 2e^−^ occurs at the anode, resulting in an alkaline pH at the interface, leading to a reduction in adhesion strength or even failure [98,99].

As mentioned in some papers, there is no difference in the magnitude of capacitance between an electrical double layer and polymer [100]. According to Equation (16), the test results would be greatly influenced by a double layer.
(41)$$\frac{1}{C_{\mathrm{total}}}=\frac{1}{C_{c}}+\frac{1}{C_{\mathrm{dl}}}=\frac{1}{C_{c}}\left(\frac{C_{c}}{C_{\mathrm{dl}}}+1\right),$$
where *C*_total_ is the capacitance of the metal–polymer system, *C*_c_ is the capacitance of the polymer, and *C*_dl_ is the capacitance of the double layer.

The corrosion process occurring at the substrate dominates the impedance response [101]. This is estimated as the ratio of capacitance of the double layer measured for the scribed systems to the value of 25 μF/cm^2^, which is the value often accepted for the double-layer capacitance of a metal/solution interface. The capacitance of the double layer was obtained using a modified equation proposed by Büchler et al. [102]. This equation accounts for the deviation from the ideal capacitive behavior, considering a constant phase element (CPE) in the equivalent circuit instead of a capacitor.
(42)$$C_{\mathrm{dl}}=\frac{1}{{\left(2\pi f_{\theta_{\max}}\right)}^{\frac{\theta_{\max}}{-90}}Z_{\mathrm{imag}}},$$
where $f_{\theta_{\max}}$ is the frequency corresponding to the maximal phase angle of impedance, *θ*_max_ is the maximal phase angle of impedance, and *Z*_imag_ is the imaginary part of the impedance at the maximal phase angle.

In order to reduce the error induced by an electrochemical reaction at the interface, some scholars proposed replacing the metal with a non-metal material with characteristics of high impedance and densification, such as PE (polyethylene) [103,104]. However, a non-metal material added into the system would increase the system’s resistance, which may result in the non-response of the test system with a regular test signal (e.g., 10 mV). On the other hand, an increase in test signal would deviate from the characteristic of EIS, potentially even causing damage to the coating. Therefore, the physical meaning and accuracy of the results obtained via the new test method should be guaranteed while reducing the influence of the interface.

The issue with the system response using a PE base is similar to the evaluation of a polymer with high resistance. According to the characteristics of EIS, Akbarinezhad et al. [105] discussed the evaluation of polymer performance under the conditions of alternating current interference. Compared to directly enlarging the test signal of EIS, this method regards the alternating current signal as an external parameter, not affecting the EIS system. However, several questions are worth discussing using this method: (1) What is the difference or relationship between results obtained using the conventional and new methods? (2) What is the physical meaning of the result from the new method? (3) Is the existing standard available to analyze the result from the new method? Furthermore, in this new method, the additional applied alternating current signal may induce a change in polymer structure and distribution of electrolyte in polymers, which may break or deviate from the balance. Therefore, for a specific system, it is crucial to ensure the critical value of alternating current signal and test cycle.

## 5. Understanding of Water Diffusion in Polymers

The basic understanding of water diffusion in polymers was tremendously advanced during the past few years, as illustrated by the examples discussed above. Although well-defined water–polymer studies can be painstaking, they are nonetheless incredibly useful as they reveal how water behaves in polymers. This is highly valuable information. Below, we discuss some of the key questions that these studies now allow us to answer, which we could only speculate on just a few years ago.

What is the ideal process of water diffusion in polymers described by Fick’s second law? There are considerable microporous diffusion channels in polymers due to curing, during which water enters the polymers due to the concentration gradient. Under this condition, the *φ* (mass fraction of water) is proportional to *t*^0.5^ (diffusion time) [17]. Once the water content in polymers reaches saturation, the diffusion flux becomes zero, and the volume fraction of water in polymers does not change over time, unless water is consumed due to electrochemical reactions at the metal/polymer interface [23].

What happens to water in polymers? It is extremely difficult for individual water molecules to exist in polymers due to their high mobility and preference to form H-bonded clusters [63]. In the case of enough water molecules in polymers, individual water molecules prefer to coalesce to form water clusters, which reduces the free volume occupied, thereby inducing more water uptake. Moreover, there are many functional groups in polymers, possibly condensed by water, leading to a change in the existing state of water from free water to bound water [59,64,65,66,67,68,69,70,71].

What are the main factors governing the exfoliation of metal and polymer? Water extends along the interface to form a layer of water molecules once water reaches the polymer/metal interface, albeit not causing a loss of wet adhesion [96,97]. The main reason for polymer delamination is an electrochemical reaction at the polymer/metal interface (Fe→Fe^2+^ + 2e^−^ at the anode and O_2_ + 2H_2_O + 4e^−^→4OH^−^ at the cathode) [98,99,106], during which the aggregation of corrosion products and the alkalization at the interface disrupt the bond between the polymer and metal, leading to a loss of wet adhesion and subsequent delamination.

Beyond the questions we currently have credible answers to, there are many more that we cannot yet answer with confidence. Some particular questions that investigators are currently focusing on after studying them for many years are listed below.

What are the conditions under which water clusters or an interaction between water and functional groups occur? Van Westing et al. [56] introduced a swelling coefficient (SC) to modify the errors caused by changes in polymer thickness. However, even for specific polymers, the SC cannot be clearly defined. Furthermore, considerable scholars put forward many different conclusions with significant controversy, which mainly focus on the three following aspects:There is only the phenomenon of water clusters, with no interaction between water and functional groups, as proven by the fact that water can be completely desorbed from polymers.Water interacts with polymers; however, under certain conditions, such as high temperature and dry conditions, it completely desorbs.An irreversible interaction between water and functional groups occurs.

There is no evidence or acceptable theory to explain the water–polymer issue. Moreover, the solution of this problem can also answer the definition of pre-diffusion, satisfying ideal Fick’s diffusion.

How do electrochemical reactions at the metal/polymer interface evolve? Due to the occlusion of the polymer/metal interface, there is still no available method to clarify the electrochemical reaction at interface, instead relying on several assumptions. Undoubtedly, delamination is severely affected by the aggregation of corrosion products and alkalization at the interface. Furthermore, oxygen at the interface is exhausted as time goes by, thereby affecting the oxygen concentration difference batteries, where the base metal works as an anode. The hydrolysis of Fe^2+^ at the interface changes the pH from alkaline to acidic, and H^+^ is concomitantly generated. Obviously, O_2_ and occlusion play crucial parts in these reactions, and we can understand how these reactions play out as discussed. However, it would be premature to conclude that we fully understand these reactions [107] without, for example, the capacity to correctly predict the effect of H^+^ produced by the hydrolysis of Fe^2+^, which also presents a challenge for interface pretreatment or a transition layer to prevent delamination [108,109,110].

Theoretical prospective. It should be clear by now that, in this area, there is an almost symbiotic relationship between experiment and theory, with, for instance, effective capacitance often relying on EIS to revise the B–K equation. It is only appropriate then to comment on some of the challenges remaining for the theoretical approach. The gravimetric method and B–K method used in most routine water-uptake studies have various well-known limitations. One of the key limitations is the difficulty in describing diffusion processes deviating from ideal diffusion. It is encouraging that related models are now being put forward to be applied to these intricate processes, also named non-Fick’s diffusion, thereby addressing this mystery primarily based on the two following breakthroughs:Polymer relaxation, more precisely, the nature of vitreous matter, which is one of the 125 questions posed in Science in 2005.H-bonds of water–water, water–polymer, and water–metal, which are central to interpreting the changes in the polymer’s internal energy caused by water diffusion.

Based on the gravimetric and capacitance methods, the error in calculating the water diffusion coefficient in polymers was summarized. To conclude, the past few years saw tremendous progress in the fundamental understanding of water diffusion in polymers [111,112]. Important concepts in relation to non-Fick’s diffusion emerged, but there remains much to be understood in terms of further developing the basic principles that control the water–polymer interaction or water clusters, and in using these principles to improve technological processes such as better waterproof coatings or water purification.

## Figures and Tables

**Figure 1 polymers-12-00138-f001:**
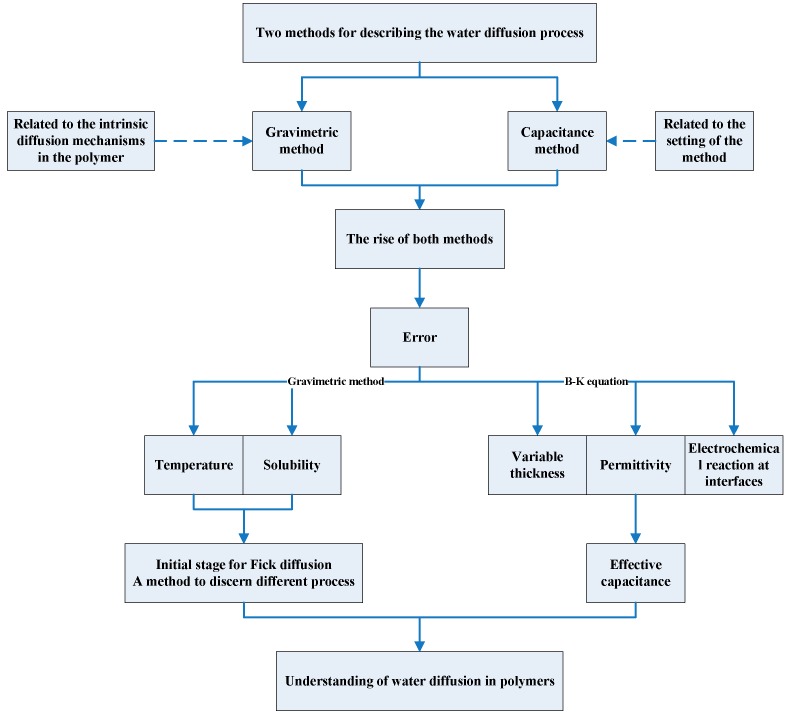
A diagram for this review.

**Figure 2 polymers-12-00138-f002:**
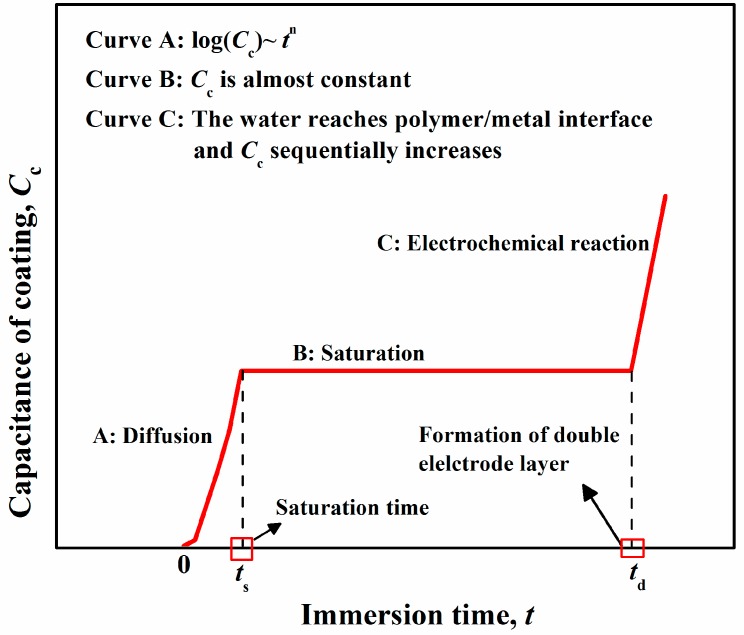
Change in polymer capacitance caused by water diffusion over time.

**Figure 3 polymers-12-00138-f003:**
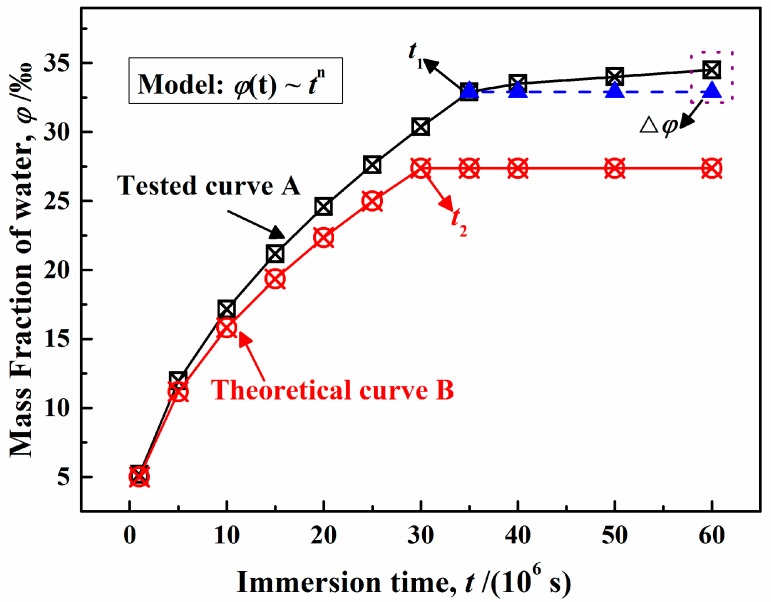
Analysis of error between experimental and theoretical values.

**Figure 4 polymers-12-00138-f004:**
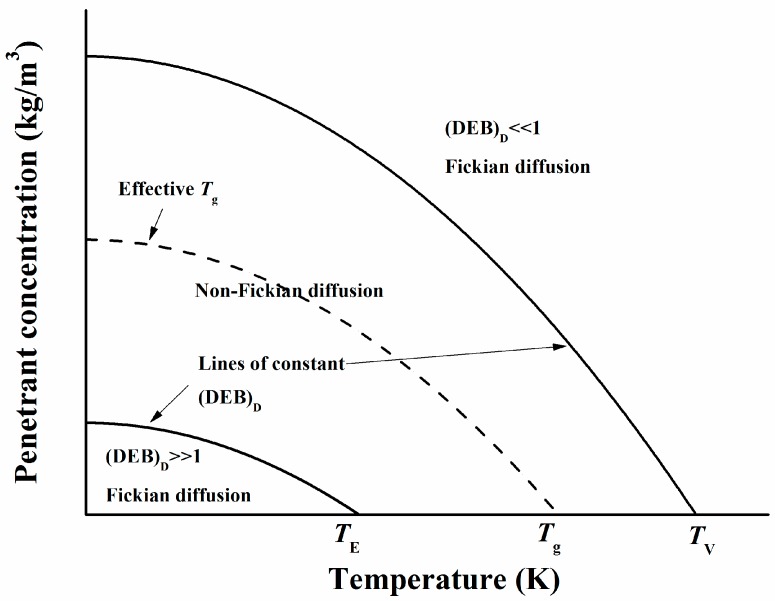
Schematic picture of the different zones of diffusion, separated by lines of constant diffusion Deborah number (DEB)_D_, as related to penetrant concentration and temperature [61].

**Figure 5 polymers-12-00138-f005:**
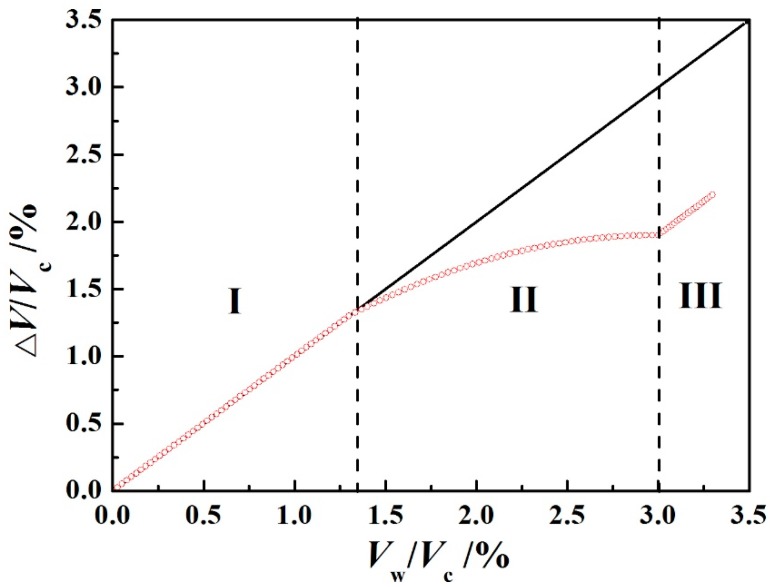
Evolution of relative swelling as a function of water uptake of free polymers at 30 °C. Δ*V* is the swelling volume of the polymer system, consisting of polymers, water, and air, *V*_w_ is the volume of water entering the polymers, and *V*_c_ is the volume of dry polymers [64].

**Figure 6 polymers-12-00138-f006:**
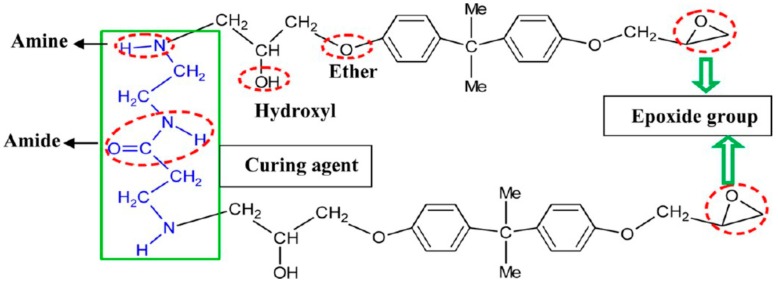
Chemical structure of aminoamide cross-linked DGEBA (Diglycidyl ether of bis phenol A) epoxy resin. The functional groups are encircled by red dotted lines.

**Figure 7 polymers-12-00138-f007:**
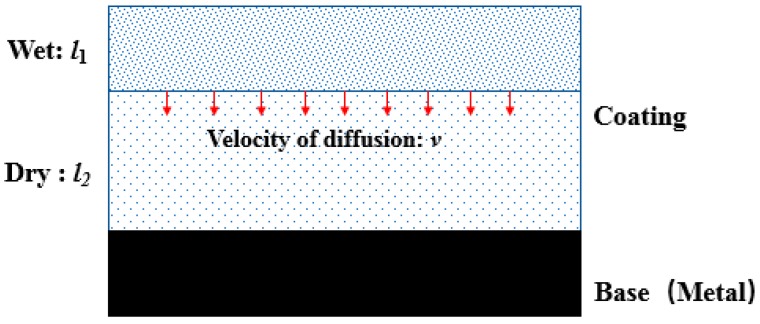
Assumption of water diffusion uniformly.

**Figure 8 polymers-12-00138-f008:**
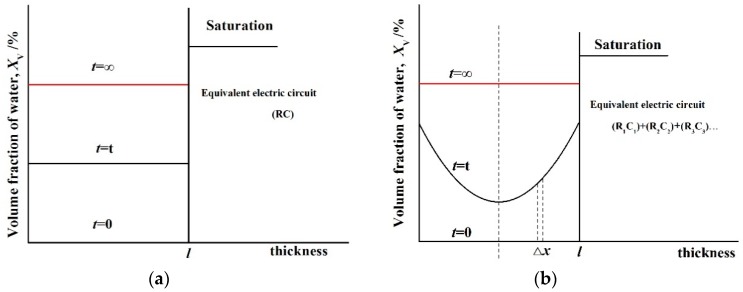
Schematic of water concentration profile in a free polymer of uniform thickness with an equivalent electric circuit according to (**a**) direct and (**b**) continuous models.

**Figure 9 polymers-12-00138-f009:**
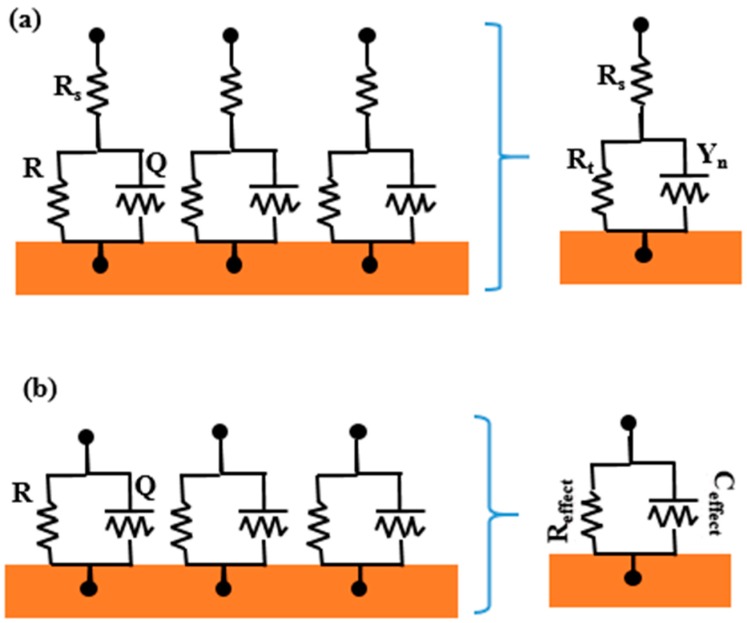
Schematic representation of a surface distribution of time constants: (**a**) distribution of time constants in the presence of an Ohm resistance resulting in a distributed time constant behavior that, for an appropriate time constant distribution, may be expressed as a constant phase angle component (CPE); (**b**) distribution of time constants in the absence of an Ohm resistance resulting in an effective RC (resistance-capacitance) behavior. The admittance *Y* shown in (**a**) includes the local interfacial and Ohm contributions [91].

**Table 1 polymers-12-00138-t001:** Water sorption characteristics assuming successive mechanisms [54].

Parameter	Immerision Temperature (°C)		
	20	40	85
Diffusion coefficient (10^−9^ cm^2^/s)	1.77 ± 0.02	11.4 ± 1.5	249 ± 35
Solubility (g/g)	0.0391 ± 0.0002	0.0342 ± 0.0002	0.274 ± 0.0016
Permeability (10^−9^ cm^2^/s)	0.0692 ± 0.0004	0.391 ± 0.053	6.76 ± 0.58

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
