# Peer review of "A Comprehensive Review on Water Diffusion in Polymers Focusing on the Polymer–Metal Interface Combination"

_polymers, 2020, doi:10.3390/polym12010138_

Round 1
Reviewer 1 Report
The article can be accepted for publication provided that the following changes are incorporated.
*The resolution of the figures should be substantially increased.
*The units of measurement are missing both in the nomenclature list and in the text.
*Equations have different formatting
*Page 31-35 are not part of the article
Author Response
Dear Reviewer:
Thanks for your careful your careful read and thoughtful comments on previous manuscript. We have taken care of the comments into consideration in preparing our revision, which has resulted in the paper clearer, more compelling, and broader. The following summarizes what we responded to the comments.
Comment 1: The resolution of the figures should be substantially increased.
Answer: The Figure 1 and Figure 2 have been polished.
Comment 2: The units of measurement are missing both in the nomenclature list and in the text.
Answer: The units of measurements in the nomenclature have been added.
Comment 3: Equations have different formatting
Answer: Thanks for your attention. All the equations have checked to ensure their accuracy.
Comment 4: Page 31-35 are not part of the article
Answer: We are so sorry for this mistake about the paper’s format, which has been pointed put by the Editor. We have revised and deleted this section.

Reviewer 2 Report
The diffusion of water in polymers is an important topic in many technical fields such as protective coatings and packaging materials. It is well known that the special physical and chemical properties of water can result in complicated diffusion mechanisms such as Non-Fickian diffusion, especially in polar and/or glassy polymers.
In the paper, a review on diffusion phenomena and mechanisms for water in polymers is given. Experimental results for the measurement of water absorption in polymers are presented, which are mainly based on the capacitance method. The discussion of these results gives indications about the present mechanisms for water diffusion.
The paper is an important contribution to this field, since it compares different approaches and models in detail and discusses possible errors in interpretation of results. However, there are some critical issues which have to be addressed.
At the beginning of the paper, an overview over the models for diffusion of water in polymers and the corresponding diffusion paths, as described in literature, should be given. Micro-pores as diffusion paths are mentioned for the first time in line 114 in the description of the capacitance method. However, there is a significant difference between the diffusion paths in polymers above glass transition temperature (generation of micro-pores by fluctuation of polymer chains) and below glass transition temperature (pre-existing micro-pores) (Literature: Mercea, Models for Diffusion in Polymers, in: Piringer, Baner, Plastic Packaging, Wiley, 2008).
The cited literature mainly considers epoxy resins, i.e. cross-linked thermosetting polymers. If the derived results especially apply to this class of polymers, this should be made clear in abstract and introduction. It is also difficult to understand, which sections and results are related to the intrinsic diffusion mechanisms in the polymer and which are related to the setting of the capacitance method. For example, the stage of electrochemical reaction in Fig. 1 relies on the metal/polymer interface, i.e. on the setting of the method.
The majority of cited literature is from protective coatings, corrosion, electrochemistry and so on, where the capacitance method seems to be state of the art. However, water diffusion in polymers also plays an important role in packaging of food and pharmaceuticals and inorganic-polymeric multilayer films for encapsulation of flexible electronics. Here, diffusion kinetics and mechanisms are typically derived from transient permeation measurements, e.g. with carrier gas methods. For example, the comparison of the lag time and other characteristic times, derived from permeation measurement, allows to determine, whether diffusion coefficients depend on concentration (Marais, Surface & Coatings Technology 201 (2006) 868–879). Relevant papers from these topics should be cited and their findings taken into consideration.
In the sections on gravimetry and capacitance methods, descriptions of the measurement methods (not only the interpretation of the experimental data) should be included, since some of the presented results rely on these methods, e.g. the stage of electrochemical reaction in Fig. 1. Furthermore, the Nyquist curves (line 337-340), the presence of ions (line 498-502) and the results on the influence of frequency (line 584-625) cannot be understood without the description of the capacitance method.
The writing style has to be improved. The text and figures contain a large number of spelling and grammar mistakes.
15: “B-K equation” should not be abbreviated in abstract.
24: What is meant by “condensation of organic matter”? Deposition from gas phase?
41: “The B-K equation is a concept that has been used to discuss water diffusion in almost all polymer”: Please give some examples and references.
61: “The water diffusion in polymer is primarily caused by concentration gradient”: This is only true, if Henry’s law of sorption is valid for water in the polymer. However, water sorption in polymers is often described by non-linear sorption isotherms (Rogers, Permeation of Gases and Vapours in Polymers. In: Comyn, Polymer Permeability, Chapman & Hall, 1985, p. 29-34) and in these cases the gradient of chemical potential instead of the concentration gradient is the true driving force of diffusion (Peterlin, Colloid and Polymer Science 263 (1985), p. 35-41).
64: On the left side of eq. 1, d^2 c should be replaced by dc.
68: The first equation should be probably c(x, 0) = 0, i.e. initial condition for all x.
71: The quantity Mt / M_inf should be defined.
74: “It is essential to note that 4l^2 should be changed to l^2 applied on free polymer”: In my opinion, equation (3) applies to boundary conditions (2) without this change, since it’s derived by Crank for a polymer of thickness 2*l, where sorption is possible on both surfaces.
95: Derivation of eq. (7) assumes that volume Vc is constant during uptake of water. According to Sykes, Corrosion Science 46 (2004) 515–517, Vc will decrease, even if the volume of the polymer remains constant. Assumptions should be made clearer.
115: Does capacitance method allow such a clear distinction between stages 1 and 2? Normally there is an asymptotic approach to saturation.
121: Please give a reference, e.g. Crank, The Mathematics of Diffusion.
134-136: What does t=0~t2 and t=0~t1 mean? The marked points correspond to t = 0?
140: Fig. 2: Please show axis scales.
160: According to Crank, the situation where rate of diffusion is equal to that of relaxation, corresponds to n with 0.5 < n < 1.
168: “This is generally accepted to be the case for polymer”: “This” refers to which fact?
172: “the Tg reflects the curing degree of polymer”: This may be true, if one considers a special class of polymers, e.g. epoxides. In comparison between different polymers, there are also other factors such as side chains which determine glass transition temperature.
214: “two assumptions must be paid attention”. Which are these two assumptions? The next sentence is meant as “If water diffusion has no influence on physical and chemical properties of polymer, then the activation energy (Ea) of diffusion process is independent of temperature”?
218: “where, D0 is a constant, representing the diffusion coefficient of water in dry polymer”. No, D0 is the theoretical diffusion coefficient in the limit T -> infinity.
221: What is an “even distribution of temperature”?
221, 224: “A fitted curve of lnD~ln(1/T)”. In an Arrhenius diagram, ln D is plotted as a function of 1/T, not ln(1/T).
228-232: Too complicated sentence
234: “the above equation relies on the gravimetry method”: Equation can also be used in other measurement methods for diffusion, e.g. carrier gas methods.
262: “The second process resulting in the deviation from ideal Fick’s law is the reaction or adsorption of water and polymer”: The sorption does not lead to a deviation from Fick’s law as long as it follows Henry’s law. A non-linear sorption isotherm gives a deviation.
275: “10-11~10-7 cm2/s“: Are these values for the permeability?
286: “at which only one process of water diffusion in polymer occurs, no permeability.” Since water vapor has to enter the polymer at the left surface according to eq. 2, there would be no diffusion without sorption. Consequently, there is also permeation.
309: “They claimed”: Who?
331: “there is only a time constant”: only one time constant
331: “resistance (Rc) decreases and capacitance (Cc) increases”: Should be made clearer, that now capacity method is again discussed. How is resistance defined in this context?
346: “different rates of simultaneous water diffusion”: There are two diffusion paths in the polymer with different diffusion coefficients? If this is dual-phase sorption and diffusion model, it should be cited here, e.g. Rogers, Permeation of Gases and Vapours in Polymers. In: Comyn, Polymer Permeability, Chapman & Hall, 1985, p. 51-52.
348-350: Relation between eq. 22 and 23 would be clearer by defining a general time constant tau in terms of the parameters of eq. 23. I guess, that tau = l^2 / D, so that tau1 and tau2 correspond to the different diffusion coefficients?
385-390: Literature reference?
428: “Delta V is the swelling volume of polymer”. I guess that it’s the difference of polymer volumes with and without water? Should be clarified in a short sentence.
471: “M_inf is the absorbed mass at infinite time”. Is this Ms, the first quantity on the right side of eq. 25?
471-473: Please give a explanation of eq. 25: Why does swelling give a contribution to Mt which increases linearly with time, e.g. does swelling generate additional channels for water transport where Fick’s law is not valid? Why does mass M0 of dry polymer contribute to mass of water?
474: Please give a link to eq. 10 which correlates mass with capacitance.
481: “alpha_1 > alpha_2 ... in Fig. 2”: Fig. 2 does not contain these quantities.
492: “the slow diffusion of ions”: Which ions are meant here?
497: Very short section. Please give a list of the three parts which will be discussed in the next sections.
512-513: “diffusion is uniform at any time, that is, the diffusion front position of water at all positions in the same level and height is the same.”: The concentration profile shows no variation along the coating area?
530-531: “water concentration into the homogeneous polymer is flat”: independent of position?
532: “is a function of time t not of the thickness of polymer”: Is this explained by the fact, that only short times are considered according to line 535?
534: Eq. 29: Literature reference? What is K? Is it correct, that phi_inf / M_inf is in eq. 29? Together with eq. 28 this means that Ct/C0 is proportional to 1/M_inf^2.
541: “divided in layers of thickness”: Which thicknesses?
566-571: Please explain the different assumptions leading to the three relationships.
570, 574, 575: Sykes instead of Skykes
578: “The denominator in the B-K equation is log(80).” Please refer to the equation, probably eq. 8.
627-630: Please give references for these statements.
644: Replace OH by OH- in first reaction equation.
647: “According to Eq.(16)”: Since thickness is increased due to double layer, this has an influence on D?
649: Eq. 40: What are C_C and C_dl (capacity of double layer)?
689: “diffusion no longer occurs”. More precise: The diffusive flux has become zero.
776: Mathematics instead of Mechanistic?
Author Response
Dear Reviewer:
Thanks for your careful your careful read and thoughtful comments on previous manuscript. We have taken care of the comments into consideration in preparing our revision, which has resulted in the paper clearer, more compelling, and broader. The following summarizes what we responded to the comments.
Question 1: At the beginning of the paper, an overview over the models for diffusion of water in polymers and the corresponding diffusion paths, as described in literature, should be given. Micro-pores as diffusion paths are mentioned for the first time in line 114 in the description of the capacitance method. However, there is a significant difference between the diffusion paths in polymers above glass transition temperature (generation of micro-pores by fluctuation of polymer chains) and below glass transition temperature (pre-existing micro-pores) (Literature: Mercea, Models for Diffusion in Polymers, in: Piringer, Baner, Plastic Packaging, Wiley, 2008).
Answer: The glass transition temperature is the key factor that affects the physical and chemical properties of the polymer. The diffusion process of water molecules is quite different in glassy and rubbery polymers. In this paper, the glass transition temperature is described in section 3.1 and 3.2, in which the error caused by the temperature in gravimetric method and capacitance method is analyzed. Lines 55~63 are added in Introduction to describe the different effects of temperature on the diffusion process. However, we believe that the main objective of this paper is to analyze the characteristics and causes of errors of the commonly used methods - gravimetric and capacitive methods, so there is no relevant diffusion model added to analyze the differences of diffusion coefficient in glass and rubber states.
Question 2: The cited literature mainly considers epoxy resins, i.e. cross-linked thermosetting polymers. If the derived results especially apply to this class of polymers, this should be made clear in abstract and introduction. It is also difficult to understand, which sections and results are related to the intrinsic diffusion mechanisms in the polymer and which are related to the setting of the capacitance method. For example, the stage of electrochemical reaction in Fig. 1 relies on the metal/polymer interface, i.e. on the setting of the method.
Answer: This paper reviews the gravimetric and capacitance method are, how it emerged, and how recent experiments and calculations reveal instead a much more interesting variety and richness of diffusion process for water in polymer.
The gravimetric method is mainly related to the properties of the polymer, and the capacitance method, derived from the mixed dielectric constant theory, is related to the electrochemical impedance spectroscopy (EIS), which has been clearly stated in Line 78~80.
Question 3: The majority of cited literature is from protective coatings, corrosion, electrochemistry and so on, where the capacitance method seems to be state of the art. However, water diffusion in polymers also plays an important role in packaging of food and pharmaceuticals and inorganic-polymeric multilayer films for encapsulation of flexible electronics. Here, diffusion kinetics and mechanisms are typically derived from transient permeation measurements, e.g. with carrier gas methods. For example, the comparison of the lag time and other characteristic times, derived from permeation measurement, allows to determine, whether diffusion coefficients depend on concentration (Marais, Surface & Coatings Technology 201 (2006) 868–879). Relevant papers from these topics should be cited and their findings taken into consideration.
Answer: The water diffusion is involved in many fields, not only in the anticorrosive coating, therefore, there are amounts of methods to determine the diffusion coefficient. However, this paper is based on gravimetric method and capacitive method, which are mainly used to describe the diffusion process of water molecules in polymer coatings. Therefore, most references cited in this paper focus on coatings instead of the others, which has been declared in Line 75~80.
Question 4: In the sections on gravimetry and capacitance methods, descriptions of the measurement methods (not only the interpretation of the experimental data) should be included, since some of the presented results rely on these methods, e.g. the stage of electrochemical reaction in Fig. 1. Furthermore, the Nyquist curves (line 337-340), the presence of ions (line 498-502) and the results on the influence of frequency (line 584-625) cannot be understood without the description of the capacitance method.
Answer: Line 132~139: The capacitance method mainly relies on electrochemical measurement, which can obtain the capacitance of whole system at different diffusion time, used in Eq.(8)~(10). A three-electrode electrochemical test system is adopted, with the coated metal as the working electrode, the saturated calomel electrode as the reference electrode and the platinum electrode as the auxiliary electrode. The electrochemical impedance spectroscopy (EIS) curves of coated metal at different diffusion times is obtained, from which the capacitance of the whole coating system is extracted by equivalent circuit fitting. It should be noted that the fitted capacitance is the average capacitance value of the whole coating system, which is greatly affected by the reaction process in the coating system.
We are so sorry for these mistakes and misleading, which has been revised point to point as follows.
15: “B-K equation” should not be abbreviated in abstract.
Revised. Brasher-Kingsbury equation.
25: What is meant by “condensation of organic matter”? Deposition from gas phase?
Revised. This includes areas such as relaxation of organic matter, electrochemistry and stress, to name just a few.
45: “The B-K equation is a concept that has been used to discuss water diffusion in almost all polymer”: Please give some examples and references.
Revised.
82: “The water diffusion in polymer is primarily caused by concentration gradient”: This is only true, if Henry’s law of sorption is valid for water in the polymer. However, water sorption in polymers is often described by non-linear sorption isotherms (Rogers, Permeation of Gases and Vapours in Polymers. In: Comyn, Polymer Permeability, Chapman & Hall, 1985, p. 29-34) and in these cases the gradient of chemical potential instead of the concentration gradient is the true driving force of diffusion (Peterlin, Colloid and Polymer Science 263 (1985), p. 35-41).
Revised. Only consider the diffusion process caused by the difference in water concentration, that is, Henry’s law of sorption is valid for water in the polymer.
86: On the left side of eq. 1, d^2 c should be replaced by dc.
Revised.
90: The first equation should be probably c(x, 0) = 0, i.e. initial condition for all x.
Revised.
96: The quantity Mt / M∞ should be defined.
Revised. Where, Mt is mass of water at time t and M∞ is the mass of water in saturation.
97: “It is essential to note that 4l^2 should be changed to l^2 applied on free polymer”: In my opinion, equation (3) applies to boundary conditions (2) without this change, since it’s derived by Crank for a polymer of thickness 2*l, where sorption is possible on both surfaces.
Revised.
115: Derivation of eq. (7) assumes that volume Vc is constant during uptake of water. According to Sykes, Corrosion Science 46 (2004) 515–517, Vc will decrease, even if the volume of the polymer remains constant. Assumptions should be made clearer.
Revised. Volume of dry polymer keeps unchanged during water diffusion.
146: Does capacitance method allow such a clear distinction between stages 1 and 2? Normally there is an asymptotic approach to saturation.
Answer: Thanks for this attention. In our opinion, the Figure 1 is only a schematic diagram of water content in different stage.
150: Please give a reference, e.g. Crank, The Mathematics of Diffusion.
Revised.
163-168: What does t=0~t2 and t=0~t1 mean? The marked points correspond to t = 0?
Answer: Yes. This section has been revised in a clearer description as follows.
Based on the assumption of ideal Fick’s diffusion, in time t=0~t2, the mass fraction of water (φ) in theory shown in curve B meets with Eq.(10), that is, n2=0.5, and once reaching the saturation state, the φ keeps unchanged. However, in time t=0~t1 shown in curve A, the φ obtained by experiment also satisfies the Eq.(10) with n1>0.5. And for the second stage of saturation, it shows a slight increase, which is different from the results in curve B.
169: Fig. 2: Please show axis scales.
Revised.
187: According to Crank, the situation where rate of diffusion is equal to that of relaxation, corresponds to n with 0.5 < n < 1.
Revised. Case II (n=1). The rate of diffusion is far faster than that of relaxation, which marks the innermost limit of water diffusion and is the boundary between (stressed equilibrium) swollen gel and the glassy core of the polymer. At this moment, swelling of polymer occurs. May be in this case, the value of n is variable, for example, in J. Crank’s opinion, Case II is defined in the range of 0<n<1 .
200: “This is generally accepted to be the case for polymer”: “This” refers to which fact?
Revised. This, rubbery state, is generally accepted to be the case for polymer above their Tg.
203: “the Tg reflects the curing degree of polymer”: This may be true, if one considers a special class of polymers, e.g. epoxides. In comparison between different polymers, there are also other factors such as side chains which determine glass transition temperature.
Revised. Actually, for another side, the Tg reflects the curing degree of polymer, especially for epoxides, which is closely related to cross-linking density and free volume.
245: “two assumptions must be paid attention”. Which are these two assumptions? The next sentence is meant as “If water diffusion has no influence on physical and chemical properties of polymer, then the activation energy (Ea) of diffusion process is independent of temperature”?
Revised. Diffusion coefficient (D) and temperature, can be theoretically illustrated by Arrhenius’s formula, in which two assumptions must be paid attention: (1) water diffusion has no influence on physical; (2) chemical properties of polymer while the activation energy (Ea) of diffusion process is independent of temperature.
250: “where, D0 is a constant, representing the diffusion coefficient of water in dry polymer”. No, D0 is the theoretical diffusion coefficient in the limit T -> infinity.
Revised.
253: What is an “even distribution of temperature”?
Revised. Uniform temperature distribution in polymer.
253, 256: “A fitted curve of lnD~ln(1/T)”. In an Arrhenius diagram, ln D is plotted as a function of 1/T, not ln(1/T).
Revised.
258-260: Too complicated sentence
Revised. On the basis of Eq.(13), there are some deformations to acquire acceptable activation energy. Eq.(14) is commonly and widely used with the need for the diffusion process to take place using the experimental values obtained for diffusion coefficient (D) at two different temperatures:
262: “the above equation relies on the gravimetry method”: Equation can also be used in other measurement methods for diffusion, e.g. carrier gas methods.
Revised. However, the above equation relies on the experimental methods, such as gravimetric method and carrier gas method, to obtain the diffusion coefficient, which undoubtedly considers the assumption of ideal Fick diffusion, especially the assumption of error in calculating diffusion coefficient.
291: “The second process resulting in the deviation from ideal Fick’s law is the reaction or adsorption of water and polymer”: The sorption does not lead to a deviation from Fick’s law as long as it follows Henry’s law. A non-linear sorption isotherm gives a deviation.
Revised. The non-linear sorption resulting in the deviation from ideal Fick’s law is the reaction or adsorption of water and polymer, which may cause the rearrangement of chemical chains, and this process is given by solubility coefficient (S).
304: “10-11~10-7 cm2/s“: Are these values for the permeability?
Answer: Diffusion coefficient, which has been revised. Much literatures suggested the order of magnitude of total “diffusion” coefficient (D and P) in different polymer was 10-11~10-7cm2/s.
315: “at which only one process of water diffusion in polymer occurs, no permeability.” Since water vapor has to enter the polymer at the left surface according to eq. 2, there would be no diffusion without sorption. Consequently, there is also permeation.
Revised. It should be noted that solubility coefficient (S) decreased with the increase of temperature, meaning that there may be an elevated temperature, at which permeation will not occur.
338: “They claimed”: Who?
Revised. J.S. Vrentas.
359: “there is only a time constant”: only one time constant
Revised.
359: “resistance (Rc) decreases and capacitance (Cc) increases”: Should be made clearer, that now capacity method is again discussed. How is resistance defined in this context?
Revised. This sentence has been deleted.
369: “different rates of simultaneous water diffusion”: There are two diffusion paths in the polymer with different diffusion coefficients? If this is dual-phase sorption and diffusion model, it should be cited here, e.g. Rogers, Permeation of Gases and Vapours in Polymers. In: Comyn, Polymer Permeability, Chapman & Hall, 1985, p. 51-52.
Revised. A phenomenological model was developed to reflect the different rates of water diffusion in more clearly established polymer due to the polymer not being a continuum medium as follows.
371-373: Relation between eq. 22 and 23 would be clearer by defining a general time constant tau in terms of the parameters of eq. 23. I guess, that tau = l^2 / D, so that tau1 and tau2 correspond to the different diffusion coefficients?
Revised. τ=l2/D in Line 377.
408-413: Literature reference?
Revised.
451: “Delta V is the swelling volume of polymer”. I guess that it’s the difference of polymer volumes with and without water? Should be clarified in a short sentence.
Revised. △V is the swelling volume of polymer system, consisted of polymer, water and air.
494: “M_inf is the absorbed mass at infinite time”. Is this Ms, the first quantity on the right side of eq. 25?
Revised. Ms→M∞.
493-496: Please give an explanation of eq. 25: Why does swelling give a contribution to Mt which increases linearly with time, e.g. does swelling generate additional channels for water transport where Fick’s law is not valid? Why does mass M0 of dry polymer contribute to mass of water?
Revised. In Figure 2, after Fick’s diffusion, there is a little increase of water content over time, which is defined in a linear relationship. The swelling, in Eq.25, is widely assumed to take place after Fick’s diffusion, which is caused by many factors, such as water clusters, ion diffusion, relaxation and so on. As for M0, we are so sorry for this mistake and have revised.
498: Please give a link to eq. 10 which correlates mass with capacitance.
Revised. Combining with Eq.(9), for the capacitance of the polymer the equivalent formula is defined as Eq.(26).
504: “alpha_1 > alpha_2 ... in Fig. 2”: Fig. 2 does not contain these quantities.
Revised. α→n.
515: “the slow diffusion of ions”: Which ions are meant here?
Answer: The ion in polymer may cause relaxation reaction to result in the swelling, especially for the OH-.
520: Very short section. Please give a list of the three parts which will be discussed in the next sections.
Revised. The error caused by permittivity in B-K equation mainly includes three parts, including ion diffusion, increasing water’s volume and individual or clustered water in polymer, as follows.
535-538: “diffusion is uniform at any time, that is, the diffusion front position of water at all positions in the same level and height is the same.”: The concentration profile shows no variation along the coating area?
Answer: In this model, it is yes, meaning that the diffusion is dependent on the thickness, which is same as the DM model.
554-555: “water concentration into the homogeneous polymer is flat”: independent of position?
Answer: In this model, it is yes.
556: “is a function of time t not of the thickness of polymer”: Is this explained by the fact, that only short times are considered according to line 560?
Answer: Yes.
556: Eq. 29: Literature reference? What is K? Is it correct, that phi_inf / M_inf is in eq. 29? Together with eq. 28 this means that Ct/C0 is proportional to 1/M_inf^2.
Revised. The equations have been checked to be OK.
566: “divided in layers of thickness”: Which thicknesses?
Revised. According to this model, the film is divided in layers of polymer’s thickness.
590-597: Please explain the different assumptions leading to the three relationships.
Answer: Considering the content of water molecules in polymers is far lower than 1, the linear relationship (Eq.32) assumed the permittivity changes linearly with the volumetric moisture content. Brasher and Kingsbury’s formula was obtained by Take the logarithmof both sides of Eq.(4). The J.M. Sykes’s formula introduced the effect of polymer on water diffusion.
596, 600, 601: Sykes instead of Skykes
Revised.
604: “The denominator in the B-K equation is log(80).” Please refer to the equation, probably eq. 8.
Revised. The denominator in the B-K equation (Eq.8) is log(80).
652-656: Please give references for these statements.
Revised.
669: Replace OH by OH- in first reaction equation.
Revised.
672: “According to Eq.(16)”: Since thickness is increased due to double layer, this has an influence on D?
Answer: In EIS test, the double layer mainly affecting the effective capacitance, which has an influence on D.
675: Eq. 40: What are C_C and C_dl (capacity of double layer)?
Revised. Cc for polymer’s capacitance and Cdl for double layer.
715: “diffusion no longer occurs”. More precise: The diffusive flux has become zero.
Revised.
812: Mathematics instead of Mechanistic?
Revised.

Reviewer 3 Report
Manuscript Title: A comprehensive review on water diffusion in polymer
This topic of review is good. However, present form of manuscript is not suitable for publication. There are several points need to be considered before this manuscript to be considered for publication.
Specific comments to authors:
Authors have to critically explain why the study of water diffusion in polymer is important? This is missing in the introduction section.Abstract should be re-framed after considering central importance of this work, what was discussed and what would be the further prospective of this research area.
What are the scopes of this review topic? This has to be pointwise discussed at the end of introduction section.
Authors are suggested to provide a flow chart at the end of introduction about the points and topics which they have covered in this review. This is important.
Effect of polymer’s characteristics on water diffusion.
There are few recent highly interesting reports on water-responsive mechano-adaptive elastomers (promising polymeric materials). There it was revealed that water diffusion in rubber matrix can develop In-Situ structure formation and can general new characteristics phenomena of the polymeric materials. These have to be included in this review (Journal of Physical Chemistry B 2019, 123, 5168 and Journal of Applied Polymer Science 2019, DOI: 10.1002/app.20191288, Temperature scanning stress relaxation behavior of water responsive and mechanically adaptive elastomer nanocomposites). Some more practical examples on water diffusion on effect of polymer’s characteristics have to be discussed. This can be a separate section. In addition to water diffusion kinetics (which authors have already discussed), technological aspects of water diffusion on physico-mechanical characteristics need to be critically discussed. This is missing in this review. More specifically, the effect of water diffusion on the physico-mechanical properties must be a separate section.
There are problems in some sub-titles. For examples, “How can we do a better job with theory?” This should be corrected as “Theoretical prospective”
“Questions we can and cannot answer” this should be corrected carefully. This could be “understand of water diffusion in polymer”. There is many more such problem in the manuscript. These have to be taken care.
A small summary and outlook of this review have to be included at the end of manuscript. This is missing.
There are many grammatical errors in the manuscript. These have to be corrected carefully.
Author Response
Dear Reviewer:
Thanks for your careful your careful read and thoughtful comments on previous manuscript. We have taken care of the comments into consideration in preparing our revision, which has resulted in the paper clearer, more compelling, and broader. The following summarizes what we responded to the comments.
Comment 1: Authors have to critically explain why the study of water diffusion in polymer is important? This is missing in the introduction section.
Answer: Thanks for your attention. The importance has been added in Introduction as follows. Indeed the ubiquitous presence of corrosive media in polymer under ambient conditions means that diffusion in polymer are relevant to many areas of the physical sciences. The study of water diffusion in polymer is widely used in the fields of analyzing the failure of coatings caused by water, and evaluating the protective performance. Furthermore, contemporary issues such as production technology and performance characterization in both self-healing coating and graphene coating mean that there is a pressing need to better understand the dynamic of water diffusion in polymer.
Comment 2: Abstract should be re-framed after considering central importance of this work, what was discussed and what would be the further prospective of this research area.
Answer: Thanks for your attention. Abstract has be re-framed. In this Review, focusing on the gravimetric and capacitance method, we discussed the contradictions and problems existing on the water diffusion in polymer in detail from the perspective of experiments and models, in particular the analysis of error derived from wildly used methods, especially for the Brasher-Kingsbury equation. We also provide a perspective on outstanding problems, challenges and open questions, including the water clusters, relaxation and electrochemical reactions at metal/polymer interfaces, and also expand the theoretical prospective.
Comment 3: What are the scopes of this review topic? This has to be pointwise discussed at the end of introduction section.
Answer: This paper is based on the gravimetric and capacitance method, to explore the cause of the error, and puts forward the further prospective, which has been emphasized at the end of introduction section.
Comment 4: Authors are suggested to provide a flow chart at the end of introduction about the points and topics which they have covered in this review. This is important.
Answer: The flow chart has been added at the end of introduction.
Comment 5: Effect of polymer’s characteristics on water diffusion.
Answer: This paper is based on the gravimetric and capacitance method, analyzing the main causes of errors in different test method. However, the effect of water molecules on polymer microstructure is not discussed in this paper. Even so, these two papers illustrate this problem so well that it is briefly illustrated in the text and used as citations.
Comment 6: There are problems in some sub-titles. For examples, “How can we do a better job with theory?” This should be corrected as “Theoretical prospective”
“Questions we can and cannot answer” this should be corrected carefully. This could be “understand of water diffusion in polymer”. There is many more such problem in the manuscript. These have to be taken care.
Answer: Thanks for your attention. They have been revised in this paper.
Comment 7: A small summary and outlook of this review have to be included at the end of manuscript. This is missing.
Answer: Thanks for the attention. The summary and outlook has been added.
Comment 8: There are many grammatical errors in the manuscript. These have to be corrected carefully.
Answer: We are so sorry for these errors and we have checked the whole paper.

Round 2
Reviewer 2 Report
The text and figures still contain some spelling and grammar mistakes, which should be corrected. Some examples:
147: Figure 1: "alomost"
159: "is too harsh" should be "are ..."
170: Figure 2: "Immesion"
250: "time" should be "temperature"
Author Response
Thanks for your reminding.
We are sorry for the spelling and grammar mistakes in the paper.
We have carefully checked and revised the full paper.
Reviewer 3 Report
Authors have addressed most of the comments carefully. Revised version can be considered for publication.
Author Response
Thanks for your review of this paper.